# Efficacy of a Smartphone Application to Promote Maternal Influenza Vaccination: A Randomized Controlled Trial

**DOI:** 10.3390/vaccines10030369

**Published:** 2022-02-27

**Authors:** Ya-Wen Chang, Shiow-Meei Tsai, Pao-Chen Lin, Fan-Hao Chou

**Affiliations:** 1Department of Nursing, National Tainan Junior College of Nursing, Tainan City 700007, Taiwan; yawen@mail.ntin.edu.tw (Y.-W.C.); amytsai@ntin.edu.tw (S.-M.T.); paochen@ntin.edu.tw (P.-C.L.); 2College of Nursing, Kaohsiung Medical University, Kaohsiung City 80708, Taiwan

**Keywords:** pregnant women, influenza vaccination, knowledge, attitudes, behavior intention, APP

## Abstract

The maternal vaccine coverage rate has been low in Taiwan. We developed an “Influenza Vaccination Reminder Application” and evaluated its efficacy in improving vaccination intention among pregnant women in Taiwan. A randomized controlled trial was conducted to compare the positive change in vaccination intention between the experimental group and the control group. Pregnant women who were more than 20 years old and at less than 32 weeks of gestation were recruited from four regional hospitals in southern Taiwan during November 2020 to April 2021. Pregnant women were randomly assigned to the experimental group, to whom the “Influenza Vaccination Reminder Application” was provided for at least two months, while pregnant women in the control group received regular maternal education only. The differences in knowledge about influenza and its vaccines, attitudes towards maternal influenza vaccination, and behavior intention of influenza vaccination among pregnant women before and after the experiment intervention were compared between two groups. The results included 126 women in the experimental group and 117 women in the control group and showed that the “Influenza Vaccination Reminder Application” increased pregnant women’s knowledge about influenza and vaccines (percentage increase in the experimental group and control group: 11.64% vs. 7.39%), strengthened their positive attitudes towards maternal influenza vaccination (percentage increase: 5.39% vs. 1.44%), and promoted positive behavioral intention toward influenza vaccination (proportion of participants with positive change in vaccination intention: 17.46% vs. 7.69%). The study supports use of “Influenza Vaccination Reminder Application” to promote the behavior intention of influenza vaccination among pregnant women in Taiwan.

## 1. Introduction

Multiple studies have reported that maternal influenza vaccination can protect pregnant women by not only effectively reducing their morbidity, hospitalization, and mortality rates [1,2,3], but also reducing the risks of low birth weight, premature birth, and stillbirth [3,4,5]. In addition, influenza vaccination is cost-effective and can prevent excessive medical burdens [6,7]. The World Health Organization (WHO) recommended that pregnant women during any trimester of their pregnancy should always receive influenza vaccination before annual influenza season for maternal-fetal health protection [8].

Research has shown that the risk of influenza vaccination side-effects among pregnant women is not different to that among the general population, and that the vaccination does not have significant adverse reactions on either the pregnant woman or the fetus [9,10]. However, clinical reports have indicated that some pregnant women remain doubtful and concerned about the safety of influenza vaccination, and therefore hold a distrustful attitude toward the vaccine [11,12,13]. Since 1998, the Taiwan Centers for Disease Control (TCDC) has directed and coordinated the annual seasonal influenza vaccination program including vaccination promotion strategies, such as public communication, increase of vaccination stations, expert advocacy, adverse event monitoring, and clinician education, etc [14]. Moreover, thanks to the implementation of National Health Insurance in Taiwan, pregnant women can receive regularly scheduled low-cost or free prenatal care, which includes maternal vaccination education. Even after the implementation of publicly funded influenza vaccination for pregnant women in 2014, the coverage rate of influenza vaccination among pregnant women in Taiwan is estimated to be as low as 30% [15].

Mobile health (mHealth) technologies are becoming commonplace to support health behavior changes [16]. For example, prior research had found that short message service reminder was only modestly effective [17]. Based on the wide use of smartphones, we developed an “Influenza Vaccination Reminder Application” (APP for short) for pregnant women to facilitate knowledge sharing, concept advocacy, and reminder services about influenza and its vaccines. This study aimed to evaluate the efficacy of the APP to promote maternal vaccination against influenza disease. The use of the APP was expected to improve the behavior intention of pregnant women to receive influenza vaccination as well as to reduce the risk of missing vaccinations, thereby effectively increasing the coverage rate of influenza vaccination among pregnant women in order to decrease maternal and infant morbidity and mortality.

## 2. Materials and Methods

### 2.1. Study Design and Population

This randomized controlled trial with ethical approval included pregnant women who received prenatal care at one of the four participating private hospitals in southern Taiwan during November 2020 to April 2021. These hospitals had large enough subject base to result in statistical stability, consented to the purpose and method of this study, and agreed to provide private designated areas for conducting the study. Permuted block design was used to allocate study participants. Randomization sequences generated with the use of a random-number table would be prepared by a researcher who was not directly involved with the study and would be kept in opaque, sealed, numbered envelopes. The permuted block size varies by hospital and it was blinded to the hospital staff. This study adopted a self-developed “Influenza Vaccination Reminder Application” as the intervention and evaluated its effects on pregnant women’s knowledge about influenza and its vaccine and attitudes towards maternal influenza vaccination to further evaluate its impact on pregnant women’s intention to influenza vaccination.

### 2.2. Study Procedure

This study adopted the experimental design and was carried out in two stages, i.e., pre- and post-tests (Figure 1).

(1)Pre-test stage

The study recruited pregnant women who met the inclusion criteria and signed the informed consent according to the principle of convenience sampling. The included subjects were subsequently randomly assigned to a control group and an experimental group, after which they were required to complete the pre-test questionnaires.

(2)Post-test stage

Subjects in the control group only received regular maternal education. Maternal education, including introduction of influenza maternal vaccination, is provided to pregnant women during regular prenatal care visits. The education materials were prepared based on the Maternal Health Booklet and Maternal Medical instructions Booklet developed by the Taiwan Ministry of Health and Welfare. In contrast, subjects in the experimental group were also invited to install and use the “Influenza Vaccination Reminder Application” (detailed below) in addition to regular maternal education. The APP would upload public announcements, real-time news, epidemic prevention policies, and public health information either periodically or aperiodically. In addition, it would remind pregnant women to get influenza vaccines and request feedback on their vaccination status every two weeks. After two months of intervention, subjects in both the control group and the experimental group were required to return to the hospital to undertake next prenatal care as well as completing the post-test questionnaires. Once the results were collected, improvements in the knowledge about influenza and its vaccine, attitudes towards maternal influenza vaccination, and influenza vaccination intention before and after the intervention were compared between the two groups.

The participants were informed about the study procedure but not the study objective. For example: the participants were informed that the pre-test and post-test questionnaires would be conducted, the contents of the questionnaires were not revealed in advance, the participants in the experimental group were only informed about the APP’s operation methods and function introduction, and the assistants who delivered the questionnaire to the participants were blinded to the treatment assignment. This would minimize the potential bias resulting from participants in the experimental group knowing the purpose of the App and responding differently to questionnaires.

### 2.3. Development of Influenza Vaccination Reminder Application

The “Influenza Vaccination Reminder Application” features a design based on the user scenario, simple operation methods, tailored services, and multiple network resources as well as push notifications. Its interface includes four major functions as below:(1)Reminder function (Home page)

The APP displays real-time news, epidemic information, and the latest government announcements from time to time, thereby updating pregnant women with the latest information and developments. In addition, every two weeks, the APP actively sends reminders of influenza vaccination through push notifications, which also include maps to vaccination centers. As a result, pregnant women can be reminded to get vaccinated on a timely basis and can easily find the locations of nearby vaccination centers.

(2)Knowledge base function

The knowledge base function of the APP provides related knowledge on influenza and its vaccine, epidemic prevention policies, consultation hotlines, and answers to common questions. In addition, it has compiled the latest news and epidemic information and divided them into nine databases, including understanding influenza, influenza vaccine, vaccination information, maternal-fetal care, physician’s column, news corner, public announcements, common questions, and consultation hotlines, thereby granting pregnant women with easy access to view, browse, and search desired contents at any time. The sources of the APP’s contents include influenza and vaccination promotion materials (such as short films, posters, brochures, animations, etc.) from the TCDC and epidemic information from the websites of various government agencies. Once access to these materials provided by government agency and professional societies after use authorizations were obtained would be evaluated and modified by two obstetricians, a public health scholar, and four authors with nursing education experience, the APP either downloads the contents or establishes a link to the website, after which they are compiled and sorted. Therefore, the APP’s contents are both credible and reliable.

(3)Calendar function

The calendar function of the APP allows pregnant women to record their to-do lists manually and sends them reminders before the due date to facilitate time management.

(4)Vaccination feedback function

The vaccination feedback function of the APP allows pregnant women to record the exact dates to get the influenza vaccine, so as to maintain information regarding the vaccination status and time of pregnant women. It also facilitates the timely termination of vaccination push notifications for those who have already been vaccinated, thereby reducing unnecessary interference.

### 2.4. Measurement of Characteristics and Outcomes

This study adopted the self-administered structured questionnaire for data collection. In addition to the basic demographic and health information, the questionnaire consists of three sections, namely knowledge about influenza and its vaccine, attitudes towards maternal influenza vaccination and influenza vaccination intention. For the content validity of these three sections, five expert scholars and practitioners in the fields of public health, clinical obstetrics, and nursing were invited to perform expert assessments. The experts reviewed the significance of the content, the clarity of the questions, and the appropriateness of wording, and provided corresponding corrections and guidance to ensure that the Content Validation Index (CVI) of the questionnaire was above 0.93. Subsequently, 30 pregnant women were invited to preliminarily test the questionnaire, and the results were used for the internal consistency and reliability analysis. The Cronbach’s α of all questions was within the acceptable 0.75–0.90 range. The questionnaire was then finalized after revisions were made based on the participants’ suggestions.

(1)Knowledge scale about influenza and its vaccine

After referencing relevant literature [18,19,20], the authors concluded that the knowledge scale could be divided into subscales covering five perspectives, i.e., “Characteristics of influenza virus” (three questions), “Severity of influenza” (four questions), “Benefits of influenza vaccination” (six questions), “Timing of influenza vaccination” (five questions), and “Safety of influenza vaccination” (three questions), adding up to a total of 22 questions. The options were “True”, “False”, or “Unknown”. A correct answer would score one point, whereas an incorrect/unknown answer would score no point. The sum of the points from all 22 questions was calculated as total knowledge scale score. The higher the knowledge scale score, the better understanding of influenza and its vaccine on the related question or field.

(2)Attitude scale towards maternal influenza vaccination

Using the Health Belief Model (HBM) proposed by Rosenstock et al. (1988) as the theoretical basis, the attitude scale of the questionnaire was a self-developed scale that investigated the attitudes of pregnant women towards maternal influenza vaccination [21]. The scale was divided into six perspectives, namely “Perceived susceptibility”, “Perceived severity”, “Perceived benefits of action”, “Perceived barriers of action”, “Cues to action”, and “Self-efficacy” [7,18,19,22,23]. Each perspective included a subscale consisting of five questions, adding up to a total of 30 questions. The questions were scored using the Likert five-point scale, with five representing “Strongly agree”, four representing “Agree”, three representing “Neutral”, two representing “Disagree”, and one representing “Strongly disagree”. The sum of the points from all 30 questions was calculated as total attitude scale score. The highest total attitude scale score was 150, while the highest score of each subscale was 25. The higher the attitude scale score, the more positive the attitude and the degree of recognition towards a certain issue or aspect.

(3)Influenza vaccination intention survey

The influenza vaccination intention was assessed directly with the participant’s willingness to receive “influenza vaccination” during pregnancy, the options of which included “Willing to be vaccinated”, “Unwilling to be vaccinated”, or “Unsure”. During the recruitment stage, the study already excluded pregnant women who had received influenza vaccination in the influenza season. Therefore, the three options of vaccination intention were re-grouped, with “Unwilling to be vaccinated” and “Unsure” categorized as “Without intention”, and “Willing to be vaccinated” as “With intention”.

### 2.5. Statistical Analysis

All valid questionnaire data collected in this study were imported into the SPSS for Windows Release 24.0 software package (SPSS Inc., Chicago, IL, USA) for descriptive and inferential statistical analysis. A difference with a *p* value less than 0.05 was considered significant. Descriptive statistics, including frequency, percentage, mean value, and standard deviation, were adopted to demonstrate the distributions of sociodemographic characteristics, knowledge scale score, attitude scale score, and vaccination intention.

In terms of inferential statistics, the chi-square test was utilized to determine the pairwise difference of sociodemographic characteristics. Alternatively, the t-test was adopted to compare the pre- and post-test knowledge/attitude scale scores of both groups. In addition, it was used to identify the percentage increase in knowledge scale score and attitude scale score owing to the intervention, of which the percentage increase in knowledge/attitude scale score is calculated as {[(post-test score) − (pre-test score)]/(pre-test score)} × 100%. Subsequently, the analysis of covariance (ANCOVA) was adopted to discuss whether post-score was significantly different between groups after controlling for pre-test score. The McNemar test was utilized to compare the differences between pre- and post-test proportions of the influenza vaccination intention within each of the experimental and control groups; the Generalized Estimating Equation (GEE) method was used to assess the effects of the APP’s intervention on influenza vaccination intention between both groups. The primary outcome was having positive change in vaccination intention and the secondary outcomes were increases in knowledge and attitude scale scores. Therefore, the multiple logistic regression analysis was utilized to explore the predictors of the positive change in vaccination intention among pregnant women. A positive change in vaccination intention was defined if the pregnant woman’s influenza vaccination intention changed from “Without intention” in the pre-test to “With intention” in the post-test. Independent variables used in the analysis included sociodemographic characteristics, group, pre-test knowledge scale score, pre-test attitude scale score, etc.

Sample size calculations were performed using Sample-Size-Calculator (Available online: https://clincalc.com/stats/samplesize.aspx (accessed on 19 October 2020)). It was assumed that the initial vaccination intention was 30%, which was raised to 50% after the intervention. Under the parameter setting of enrollment ratio = 1, α = 0.05, and power (1 − β) = 0.80, it was calculated that the required sample size was 186. With a turnover rate of approximately 50%, the estimated final sample size was 280, including 140 in the control group and 140 in the experimental group.

## 3. Results

### 3.1. Sociodemographic Characteristics

In this study, a total of 302 participants who met the inclusion criteria were recruited for the pre-test, 153 of whom were assigned to the experimental group and 149 to the control group. After two months of intervention, the effective sample size of the experimental group and the control group was 126 and 117, respectively, once participants who failed to attend the follow-up in time, finish the questionnaire, or provide complete answers were excluded. The recruitment of participants and the process of sample collection was shown in Appendix A.

Of the pregnant women in the experimental group, the average age and average pregnancy weeks were 31.98 ± 5.05 years and 23.13 ± 7.82 weeks, respectively. Alternatively, of the pregnant women in the control group, these were 32.10 ± 5.63 years and 22.14 ± 7.42 weeks, respectively. There were no significant differences between the distributions of the sociodemographic characteristics of the two groups (Table 1).

### 3.2. Knowledge about Influenza and Its Vaccine

The pre-test knowledge scale scores about influenza and its vaccine are listed in Table 2. The overall correct answer rate was approximately 62% (64% in the experimental group vs. 60% in the control group). Analysis of the subscales on the five perspectives of the knowledge scale indicated that “Safety of influenza vaccination” had the lowest correct answer rate of 43%, while that of the remaining four perspectives was around 60%. In addition, there were no significant differences between the pre-test knowledge scale and subscale scores of the two groups (*p* ≥ 0.05).

Comparison between the pre- and post-test total knowledge scale scores of the two groups (Appendix A) indicated that the percentage increase of the experimental group was 11.64% (*p* < 0.01), whereas that of the control group was merely 7.39% (*p* < 0.01). Furthermore, ANCOVA suggested that the post-test scores of the two groups were significantly different after controlling for pre-test score (*p* = 0.01). Analysis of the subscale results in Appendix A showed that for the experimental group, the post-test scores of all subscales were consistently better than the corresponding pre-test scores, with “Safety of influenza vaccination” having the largest percentage increase of 31.88%. In contrast, for the control group, the percentage increases of all subscales were less than 10%.

### 3.3. Attitudes towards Maternal Influenza Vaccination

The pre-test attitude scale scores towards maternal influenza vaccination are listed in Appendix A. The overall mean of total attitude scale score was 99.88 (99.40 in the experimental group vs. 100.38 in the control group). Of the six perspectives of the attitude scale, “Perceived severity” had the highest mean of attitude subscale score at 18.74, whereas “Self-efficacy” had the lowest mean of attitude subscale score at 14.76. There were no significant differences between the pre-test attitude scale and subscale scores of the two groups (*p* ≥ 0.05).

Comparison between the pre- and post-test attitude scale scores of the two groups (Table 3) indicated that the percentage increase in the experimental group was 5.39% (*p* < 0.01), whereas that in the control group was merely 1.44% (*p* < 0.01). Furthermore, ANCOVA suggested that the post-test scores of the two groups were significantly different after controlling for pre-test score (*p* = 0.01). The subscale results from Appendix A indicated that the experimental group showed considerable improvements in the four perspectives of “Perceived benefits of action”, “Perceived barriers of action”, “Cues to action”, and “Self-efficacy”, with the percentage increase in “Self-efficacy” being the highest at 8.86%. In contrast, the control group only showed significant improvements in the two perspectives of “Perceived susceptibility” and “Perceived severity”, and the percentage increase was only about 2%. In addition, for both the experimental and the control groups, the post-test score of “Self-efficacy” remained lower than that of the other perspectives.

### 3.4. Influenza Vaccination Intention

Table 2 shows that the pre- and post-test influenza vaccination intentions in the experimental group were significantly different (*p* < 0.01), whereas those in the control group remained unchanged (*p* = 0.27).

Prior to the introduction of the intervention, the proportions of the two groups of pregnant women who were willing to be vaccinated were both similar and homogeneous (36.51% vs. 30.77%), the difference of which was insignificant (*p* = 0.34). After the intervention, this proportion increased to 52.38% in the experimental group and 35.04% in the control group, the difference of which was significant (*p* = 0.01), as shown in Figure 2 and Table 3. The results of GEE analysis in Table 3 showed that the odds ratio of the experimental group from pre-test to post-test was 1.58 compared with the control group, and there was a significant difference (*p* = 0.01), so the APP’s intervention had a significant effect on the positive change in vaccination intention.

### 3.5. Effect of the Intervention on Positive Change in Vaccination Intention

Table 2 shows that among all subjects, 31 demonstrated a positive change in vaccination intention. Among them, 22 were from the experimental group, accounting for 17.46% of the group, and the remaining 9 were from the control group, accounting for 7.69% of that group. Thus, the proportion of participants who showed a positive change in vaccination intention in the experimental group was significantly larger than that in the control group.

In this study, subjects were randomly assigned, and there were no significant differences between sociodemographic characteristics variables of the two groups. Therefore, when analyzing factors related to the positive change in vaccination intention, only three independent variables, i.e., “Group”, “Pre-test knowledge scale score”, and “Pre-test attitude scale score”, were included in the regression model, as shown in Table 4. The results suggested that the primary influencing factor was “Group”, as the odds of a pregnant woman in the experimental group who would experience a positive change in vaccination intention was 2.41 times that of a pregnant woman in the control group (OR = 2.41, 95% CI: 1.04–5.55, *p* = 0.03). This section may be divided by subheadings. It should provide a concise and precise description of the experimental results, their interpretation, as well as the experimental conclusions that can be drawn.

## 4. Discussion

This study developed the first application in Taiwan that is targeted at pregnant women for knowledge promotion, concept advocacy, and reminder services about influenza and its vaccines, thereby improving the behavioral intention of pregnant women to receive influenza vaccination as well as reducing the risk of missing vaccinations. The intervention using the “Influenza Vaccination Reminder Application” significantly increased pregnant women’s positive behavioral intention towards influenza vaccination.

Analysis of the pre-test knowledge scale score showed that the correct answer rates for “Characteristics of influenza virus”, “Severity of influenza”, “Benefits of influenza vaccination”, and “Timing of influenza vaccination” were around 60%, which indicated that most pregnant women had some general knowledge in these perspectives. However, the correct answer rate for “Safety of influenza vaccination” was merely 40%, which reflected pregnant women’s poor understanding of this topic. Several studies have reported that the safety and effectiveness of the vaccine are important factors affecting the behavioral intention of pregnant women to receive influenza vaccination [12,20,24,25,26,27]. After two months of intervention, the percentage increase in knowledge scale score in the experimental and control groups was 11.64% and 7.39%, respectively, suggesting that the “Influenza Vaccination Reminder Application” was substantially more effective than regular maternal education in improving participants’ knowledge about influenza and its vaccine. In particular, the percentage increase in the experimental group for “Safety of influenza vaccination” was the highest, reaching beyond 30%, whilst that in the control group was insignificant. The higher pregnant women’s correct answer rate of knowledge about influenza and its vaccine, the more likely they are to receive influenza vaccination [12,19,24]; this is especially true about those who have a decent understanding of the safety and effectiveness of the vaccine [12,24,25,26,27]. Literature review suggested that addressing common misunderstandings about vaccination; promoting its benefits; and providing accurate, effective, and individualized advice that could help pregnant women achieve a solid understanding about vaccination could effectively increase their influenza vaccination rate [28,29]. Meanwhile, the use of mHealth mobile APPs as an intervention measure could productively increase users’ knowledge and establish correct health concepts [30].

Analysis of the pre-test attitude scale score based on the HBM showed that “Perceived severity” had the highest score, whereas “Self-efficacy” had the lowest score. Therefore, it was concluded that most pregnant women agreed that influenza infection would demonstrate a certain degree of serious impacts on the physical and mental health of both the mother and the fetus. However, they were hesitant or unwilling to receive influenza vaccination due to limitations in their abilities and confidence. After two months of intervention, the percentage increase in attitude scale score in the experimental and control groups was 5.39 and 1.44%, respectively, which suggested that the “Influenza Vaccination Reminder Application” was substantially more effective than regular maternal education in improving participants’ attitudes towards maternal influenza vaccination. Pregnant women in the experimental group showed the largest percentage increase in “Self-efficacy”, although its post-test score remained the lowest. Those in the control group only showed significant improvements in “Perceived susceptibility” and “Perceived severity”, and the percentage increases were minor (2%). Relevant literature has confirmed that the more positive pregnant women’s attitudes towards vaccination, the more likely they are to be vaccinated [25]. Since participants in this study scored lower in “Self-efficacy” on the attitude scale, promotion should focus on strengthening pregnant women’s confidence in the safety and quality of publicly funded influenza vaccines [28,31,32], dispelling negative or fake news through instant news and push notifications [31,33,34,35], correcting the misunderstanding that the risk of influenza infection is low and that self-protection is all what is needed to prevent infection [31,32,36], and explaining differences between vaccination side effects and adverse events as well as the fact that their risks are very small [23,37], so as to maximize the efficacy of the “Influenza Vaccination Reminder Application”.

The survey results of this study showed that about 30% of the pregnant women in Taiwan were willing to receive influenza vaccination, which was consistent with the estimated vaccination coverage rate of pregnant women from the literature [15]. After two months of intervention, the proportion of participants who showed a positive change in vaccination intention in the experimental group was significantly larger than that in the control group (17.46% vs. 7.69%). In addition, regression analysis indicated that the odds of a pregnant woman in the experimental group experiencing a positive change in vaccination intention were 2.41 times that of a pregnant woman in the control group, thereby clearly demonstrating the effect of the “Influenza Vaccination Reminder Application” in increasing pregnant women’s influenza vaccination intention. It is possible that in countries with limited access to care or in areas with scarce medical resources, the effect of APP in promoting vaccination intention might be more significant. Relevant literature previously introduced interventions using portal messages or call services to promote influenza vaccination among unvaccinated adults [38]; this result showed that for those who underwent the intervention, whether they were only receiving messages, calls, or both messages and calls, they were always more willing to receive influenza vaccination than those who only received regular health education [38]. In addition, systematic review found that the use of mHealth mobile APPs as interventions during pregnancy could promote the positive development of beliefs, health behaviors, and health outcomes [30]. Their positive effect on preventive health behaviors was more obvious [39], not to mention that interventions using mobile APPs were cost-effective [40].

## 5. Conclusions

This study showed that the “Influenza Vaccination Reminder Application” could increase pregnant women’s knowledge about influenza and its vaccine, strengthen their attitudes towards maternal influenza vaccination, and further promote their influenza vaccination intention. Accessed through smartphones that were both portable and popular, the APP could provide pregnant women with unlimited access to valuable information, promoting vaccination intention and preventing a missed opportunity to vaccinate.

## 6. Study Limitations

There were five potential limitations in the study. First, participants were recruited from selected hospitals or clinics in southern Taiwan, affecting generalizability of study findings. Second, the frequency of use of the application by pregnant women in the experimental group could not be determined, which might affect interpretation of study findings. Third, although the participants would be totally blinded as far as possible, Hawthorne effect could not be completely eliminated. Fourth, using per-protocol analysis could potentially introduce bias due to differential attrition, comparing to using intention-to-treat analysis. However, this bias could be greatly reduced because the attrition rates in two groups were similar and the reasons of attrition were not different. The last potential limitation was the possibility of “contamination” between two groups as participants in the experimental group might share App installation with participants in the control group. This could potentially attenuate the effect of the App.

## Figures and Tables

**Figure 1 vaccines-10-00369-f001:**
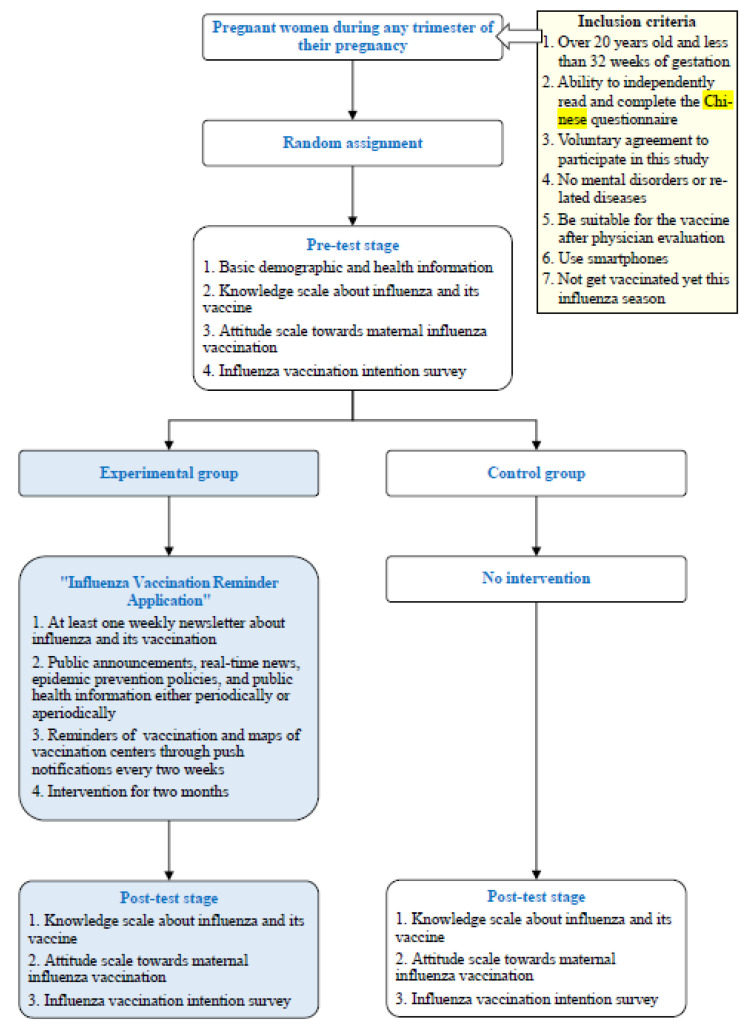
Study procedure.

**Figure 2 vaccines-10-00369-f002:**
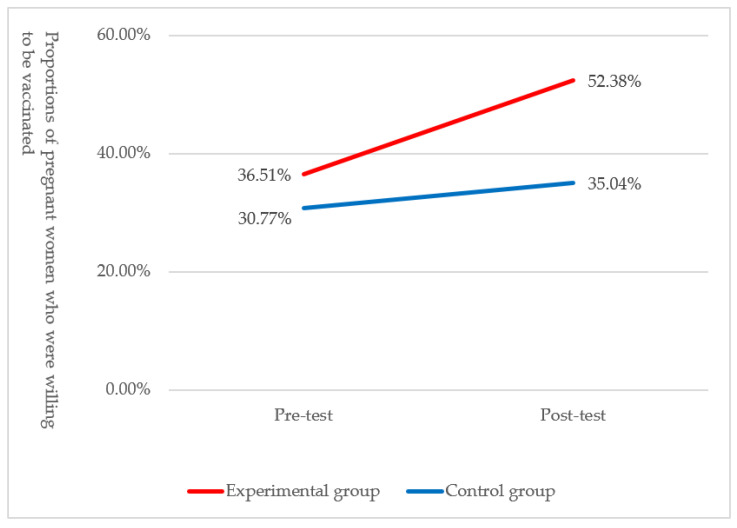
Change in the pre- and post-test proportions of the influenza vaccination intention between two groups of pregnant women.

**Table 1 vaccines-10-00369-t001:** Description of sociodemographic characteristics between two groups of pregnant women.

Sociodemographic Variables	All (*n* = 243)	Experimental Group (*n* = 126)	Control Group (*n* = 117)		
	*n* (%)	*n* (%)	*n* (%)	χ^2 a^	*p*
**Age group**				0.35	0.56
<34 years	150 (61.73)	80 (63.49)	70 (59.83)		
≥34 years	93 (38.27)	46 (36.51)	47 (40.17)		
**Trimester of pregnancy**				6.04	0.05
1st trimester	34 (13.99)	20 (15.87)	14 (11.97)		
2nd trimester	81 (33.33)	33 (26.19)	48 (41.03)		
3rd trimester	128 (52.68)	73 (57.94)	55 (47.00)		
**Marital status**				0.17	0.68
Unmarried	15 (6.17)	7 (5.56)	8 (6.84)		
Married	228 (93.83)	119 (94.44)	109 (93.16)		
**Highest education**				2.87	0.09
≤High school	55 (22.63)	23 (18.25)	32 (27.35)		
≥Junior college	188 (77.37)	103 (81.75)	85 (72.65)		
**Employment status**				0.03	0.86
Unemployed	51 (20.99)	27 (21.43)	24 (20.51)		
Employed	192 (79.01)	99 (78.57)	93 (79.49)		
**History of illness**				3.25	0.07
No	215 (88.48)	107 (84.92)	108 (92.31)		
Yes	28 (11.52)	19 (15.08)	9 (7.69)		
**Gestational complication**				0.21	0.65
No	231 (95.06)	119 (94.44)	112 (95.73)		
Yes	12 (4.94)	7 (5.56)	5 (4.27)		

^a^ Chi-square test.

**Table 2 vaccines-10-00369-t002:** The pre- and post-test proportions of the influenza vaccination intention within each group of participants.

	Post-Test	Experimental Group (*n* = 126)				Control Group (*n* = 117)			
Pre-Test		With Intention (%) ^a^	Without Intention (%) ^a^	Total (%) ^a^	*p* ^b^	With Intention (%) ^c^	Without Intention (%) ^c^	Total (%) ^c^	*p* ^b^
**With**		44	2	46	<0.01	32	4	36	0.27
**intention**		(34.92)	(1.59)	(36.51)		(27.35)	(3.42)	(30.77)	
**Without**		22	58	80		9	72	81	
**intention**		(17.46)	(46.03)	(63.49)		(7.69)	(61.54)	(69.23)	
**Total**		66	60	126		41	76	117	
		(52.38)	(47.62)			(35.04)	(64.96)		

^a^ Proportions of the influenza vaccination intention in the experimental group. ^b^ McNemar test compared the differences between pre- and post-test proportions of influenza vaccination intention within each of the experimental and control groups. ^c^ Proportions of the influenza vaccination intention in the control group.

**Table 3 vaccines-10-00369-t003:** The pre- and post-test proportions of pregnant women who were willing to be vaccinated between both groups.

	Experimental Group (*n* = 126)	Control Group (*n* = 117)	*p* ^a^	Odds Ratio ^b^
Pre-test, *n* (% ^c^)	46 (36.51)	36 (30.77)	0.34	
Post-test, *n* (% ^c^)	66 (52.38)	41 (35.04)	0.01	
*p* ^d^	<0.01	0.27		1.58

^a^ Chi-Square test compared the pre/post-test proportions of influenza vaccination intention between the experimental and control groups. ^b^ Odds ratio for Group × Pre/Post-test form GEE (*p* = 0.01). ^c^ Proportions of the influenza vaccination intention in each group of participants. ^d^ McNemar test compared the differences between pre- and post-test proportions of influenza vaccination intention within each of the experimental and control groups.

**Table 4 vaccines-10-00369-t004:** Multiple logistic regression analysis for positive change in vaccination intention among pregnant women.

Predictive Variables	*n*	Unadjusted Odds Ratio			Adjusted Odds Ratio		
		OR	95% CI	*p*	aOR	95% CI	*p*
**Group**				----			**----**
Control group	117	(Ref)			(Ref)		
Experimental group	126	2.54	1.12~5.77	**0.03**	2.41	1.04~5.55	**0.03**
**Pre-test total knowledge score**				0.32			0.43
≥15	113	(Ref)			(Ref)		
8~14	112	0.50	0.06~4.06	0.51	0.45	0.05~3.90	0.47
≤7	18	1.61	0.74~3.53	0.23	1.44	0.64~3.22	0.38
**Pre-test total attitude score**				0.10			0.11
≥120	17	(Ref)			(Ref)		
88~119	181	4.57	0.54~38.82	0.16	4.28	0.48~38.11	0.19
≤87	45	1.99	0.25~15.80	0.52	1.78	0.22~14.62	0.59
Omnibus test					χ^2^	*df*	*p*
					11.63	5	0.04
Hosmer & Lemeshow goodness-of-fit test					χ^2^	*df*	*p*
					3.97	7	0.78

## Data Availability

Not applicable.

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
