# Peer review of "Efficacy of a Smartphone Application to Promote Maternal Influenza Vaccination: A Randomized Controlled Trial"

_vaccines, 2022, doi:10.3390/vaccines10030369_

Round 1
Reviewer 1 Report
Thank you for your intersting manuscript describing the use of mobile technology to improve vaccine intent. I have a few comments and suggestions that I hope you find helpful.
- Introduction - can you describe the influenza maternal vaccination strategy in Taiwan. You could also describe other countries using mobile technology for example the UK (https://www.theholderlab.com/matimms)
- Methods - how were the hospitals selected? More detail on the app would be great - is there a link to follow to see it for other countries wishing to adopt/adapt? A description of what regular education includes would be helpful. Inclusion criteria are missing. What did you do about literacy and language? How was information to go onto the app selected and by whom? Were women involved in its design?
- Results - do you need all the tables/ it is quite dense and you could perhaps move table 3 to supplemental data as you use this table for the correlations I believe.
- Discussion - you do not include limitations to your study, such as selection bias, recall bias etc.
Author Response
Please see the attachment. Thank for your comments.

Reviewer 2 Report
This study aims to evaluate the efficacy of "Influenza Vaccination Reminder Application" to promote maternal vaccination against influenza. The manuscript is largely well-described, but the following, I think, should be improved before publication
Introduction
- Line 51-54: Although the authors present the purpose of the APP, they do not present that of the study. It should be presented here.
Methods
- What were the primary and secondary (or explorative) outcomes? Rigorous statistical significance can only be claimed for primary outcome. According to the abstract, primary outcome appears to be “the increase in the proportion of women who have an intention of receiving vaccination.” Other measures would be secondary or explorative.
- Related to #2, where was this study protocol registered? The outcomes and statistical plans should be pre-specified in the protocol, which should be registered before conducting the study. This is important to secure the transparency of the study.
- What was the randomization process? Was it complete randomization, permutated block design, or anything else? Did it involve a centralized online system, envelopes or anything else?
- I would like to ask a few questions with regard to blinding.
> How much blinded were the participants to the treatment and to the outcomes?
> Related to this, what information was provided to the participants when they were recruited?
If the investigators disclosed none of the above at the recruitment, this trial would essentially be a placebo-controlled RCT. If, on the other hand, the investigators explained something like “this study aims to evaluate whether "Influenza Vaccination Reminder Application" promotes vaccination intention,” then the participants would totally be unblinded. If there was any lack of masking, the potential limitations should be discussed. - I also would like to ask several questions as to how much comparable the control was to treatment.
> What was the “regular maternal education?”
> Was it a pre-specified, standardized one?
> Does it involve influenza vaccination?
- Statistical analysis, line 182-187: This analysis is inappropriate because it does not account for “with-to-without” conversion and because it implies “with-to-with” as negative response. You might fit a linear mixed model for the probability of “with intention” with fixed effects of pre/post, groups, and their interaction and with random effects of the individuals. In this case, the treatment effect is represented by the interaction term. Alternatively, you might apply “two-sample McNemar test.”
- Line 189: “vaccination rate”---> “vaccination intention ?”
Results
- Figure 1 should include the number of individuals at each stage and should start at eligible women who were recruited for entry. Then the refusal and exclusion follow before randomization. Please follow the CONSORT statement.
- Also, the analysis should be “intention-to-treat,” at least for primary analysis. That is, the analysis should include “dropped-out” participants, so that the subjects should consist of 302 persons instead of 243. The principle of intention-to-treat analysis is fundamental to RCTs.
- Table 2: The paired t-tests here appear to compare between pre and post scores within each of the experimental and control groups, not between.
Discussion
The authors should discuss limitations and generalizability.
Author Response
Please see the attachment. Thank four your comments.

Round 2
Reviewer 1 Report
My comments have been addressed
Author Response
Dear Mr./Ms. Reviewer:
I have benefited a lot from your suggestions and comments, thank you very much.
(PS: These additional articles [13-1] and [15-1] have been classified [14] and [17], other articles NO. have also been corrected.)
Kind Regards,
Ya-Wen Chang
Reviewer 2 Report
Thank you for taking time to revise the manuscript. I think most of my concerns have been addressed. I would like to make several additional comments below:
- Because per-protocol analysis could introduce bias due to differential attrition, employing it instead of ITT is a limitation. Accordingly, you should raise it in the limitation section. Do the rates and reasons of attrition in both groups support the absence of such bias?
- Response 5, “in order to reduce performance bias, the participants were informed that the pre-test and post-test questionnaires would be conducted. The contents of the questionnaires were not revealed in advance. The participants in the experimental group were informed about the APP’s operation methods and function introduction. This would reduce the bias by knowing the purpose of the study.”: I could not not understand these sentences (what “performance bias” is; what “This” represents and why it would reduce the bias by knowing the purpose of the study, etc.). Perhaps, you can explain more details in the methods section so that you can only use a brief summative phrase here.
Also, you should describe that the interviewers were blinded to the treatment assignment. (I am sorry to forget to mention it in the previous round.) - Another potential limitation includes the possibility of “contamination” among treated and control participants because they were recruited sequentially in the same hospitals. Do you have any good reasons that treated participants did not pass the APP-provided information to control ones (during maternal education program held in the same hospitals, for example)?
- Line 70-73: You should mention what the block size was and that it was blinded to the hospital staff. (Unblinded block size, especially small ones, could throw the investigators into the situation that enables manipulating the entry.)
- Line 50-51 “substantially lower than the 80% coverage rate required for herd immunity”: I am sorry to point it out at this stage, but it sounds strange to compare maternal vaccination rate with the threshold for herd immunity because the primary aim of influenza vaccination among pregnant women is not obtaining herd immunity but protecting maternal and fetal health.
Author Response
Dear MR./Ms. Reviewer:
I have benefited a lot from your suggestions and comments, I have revised manuscript as the attachment. Thank you very much.
Kind Regards,
Ya-Wen Chang
